# Inhibition of TLR4 Alleviates Heat Stroke-Induced Cardiomyocyte Injury by Down-Regulating Inflammation and Ferroptosis

**DOI:** 10.3390/molecules28052297

**Published:** 2023-03-01

**Authors:** Dandan Chen, Yao Geng, Ziwei Deng, Peiling Li, Shujing Xue, Tao Xu, Guanghua Li

**Affiliations:** 1Department of Physiology, Basic Medical School, Ningxia Medical University, Yinchuan 750004, China; 2School of Nursing, Ningxia Medical University, Yinchuan 750004, China; 3School of Public Health and Management, Ningxia Medical University, Yinchuan 750004, China

**Keywords:** HS, TLR4, inflammatory response, ferroptosis, cardiomyocytes

## Abstract

Inflammatory response and cell death play key roles in the mechanism of myocardial cell injury induced by heat stroke (HS) in rats. Ferroptosis is a newly discovered regulatory type of cell death, which is involved in the occurrence and development of various cardiovascular diseases. However, the role of ferroptosis in the mechanism of cardiomyocyte injury caused by HS remains to be clarified. The purpose of this study was to investigate the role and potential mechanism of Toll-like receptor 4 (TLR4) in cardiomyocyte inflammation and ferroptosis under HS conditions at the cellular level. The HS cell model was established by exposing H9C2 cells at 43 °C for 2 h and then recovering at 37 °C for 3 h. The association between HS and ferroptosis was investigated by adding the ferroptosis inhibitor, liproxstatin-1, and the ferroptosis inducer, erastin. The results show that the expressions of ferroptosis-related proteins recombinant solute carrier family 7 member 11 (SLC7A11) and glutathione peroxidase 4 (GPX4) were decreased, the contents of glutathione (GSH) were decreased, and the contents of malondialdehyde (MDA), reactive oxygen species (ROS), and Fe_2+_ were increased in H9C2 cells in the HS group. Moreover, the mitochondria of the HS group became smaller and the membrane density increased. These changes were consistent with the effects of erastin on H9C2 cells and were reversed with liproxstatin-1. The addition of TLR4 inhibitor TAK-242 or NF-κB inhibitor PDTC reduced the expressions of NF-κB and p53, increased the expressions of SLC7A11 and GPX4, reduced the contents of TNF-α, IL-6 and IL-1β, increased the content of GSH and reduced MDA, ROS, and Fe_2+_ levels in H9C2 cells under the HS condition. TAK-242 may improve the mitochondrial shrinkage and membrane density of H9C2 cells induced by HS. In conclusion, this study illustrated that inhibition of the TLR4/NF-κB signaling pathway can regulate the inflammatory response and ferroptosis induced by HS, which provides new information and a theoretical basis for the basic research and clinical treatment of cardiovascular injuries caused by HS.

## 1. Introduction

With annual increases in global temperatures and the continuous occurrence of extremely high temperatures in summer, the incidence and mortality of HS have shown a significant upward trend worldwide [1], representing a serious challenge to public health. The cardiovascular system is involved in the whole-body heat dissipation reaction and organ perfusion during HS. Among patients with multiple organ dysfunction, the incidence of cardiovascular dysfunction can be as high as 43.4~65.2% [2]. The heart is vulnerable to damage in HS, with manifestations of arrhythmia, heart failure, and focal myocardial necrosis [3]. Studies have focused on the prevention and treatment of HS. However, while they have shown that HS can cause a variety of toxic effects on the cardiovascular system, such as inflammation, oxidative stress, and apoptosis [4,5], its complete pathogenesis remains unclear.

Ferroptosis is a regulatory form of cell death that occurs due to an excessive accumulation of iron-dependent lipid ROS in cells [6]. Studies have found that ferroptosis also plays an important role in cardiovascular diseases, such as cardiomyopathy, myocardial infarction, heart failure, coronary atherosclerosis, myocardial ischemia/reperfusion injury, and other processes of cardiovascular diseases in which ferroptosis is involved [7]. HS can lead to an imbalance in oxidation-antioxidant levels, which eventually leads to heart failure and focal myocardial necrosis. Mitochondrial shrinkage, and membrane density increase are the key characteristics of ferroptosis. In this process, it has been reported that the decrease in antioxidant capacity is mainly related to decreased SLC7A11 expression, GSH synthesis, and GPX4 activity, and increased antioxidant capacity in terms of increased lipid ROS and MDA content [6]. These findings suggest that ferroptosis may be involved in the process of HS, but the specific mechanism remains to be explored.

As a member of the Toll-like receptor family, TLR4 was the earliest toll-like receptor discovered and has the highest expression level in cardiomyocytes [8]. Various studies have shown that TLR4 plays an important role in myocarditis, myocardial ischemia, myocardial infarction, and other myocardial injuries [9]. It has been confirmed that HS can induce a systemic inflammatory response and promote the production of TNF-α, IL-6, IL-1β and other inflammatory factors, and the TLR4/NF-κB signaling pathway is also involved in mediating the inflammatory response of HS rats [10,11]. Our previous study found that ferroptosis may be involved in the process of cardiac function injury caused by HS in rats, and that inhibition of TLR4 can improve cardiac function abnormalities [12]. Studies have shown that p53 mediates apoptosis, autophagy, and cell death after the involvement of NF-κB [13] and down-regulates the expression of SLC7A11 to induce ferroptosis [14]. Given this research context, we aimed to clarify whether HS can induce ferroptosis and to investigate whether inhibition of TLR4 can reduce the inflammatory response and ferroptosis of cardiomyocytes. Therefore, in this study, H9C2 cells were selected for high-temperature exposure to investigate the effect and potential mechanisms of TLR4 on HS-induced inflammatory response and ferroptosis. The results are intended to help provide a new target and theoretical basis for the clinical treatment of HS.

## 2. Results 

### 2.1. Cardiomyocyte Injury Can Be Induced by HS in a Time- and Temperature-Dependent Manner

To explore the appropriate heat exposure temperature and time, we placed H9C2 cells in incubators at different temperatures (41 °C, 43 °C, 45°C, and 37 °C for the control group) for 2 h to observe changes in cell morphology and cell viability at different temperatures. The results showed that the H9C2 cells at 37 °C showed a long spindle shape. Compared with the case at 37 °C, the cytoskeletal structure gradually began to change and began to wrinkle at 41 °C. At 43 °C, the H9C2 cells gradually shrank and showed an irregular shape. At 45 °C, the H9C2 cells showed an irregular balloon shape (Figure 1A). The CCK8 results showed that cell viability decreased gradually with the increase in the intervention temperature within a certain period of time (*p* < 0.01) (Figure 1B). Further, we placed H9C2 cells at 43 °C for a different duration of intervention, and the results showed that cell viability decreased gradually (*p* < 0.05, *p* < 0.01) (Figure 1C) and cell morphology began to change from a long spindle shape to an irregular oval shape (Figure 1D) with the increase in intervention time. Therefore, heat stroke was shown to be able to cause changes in H9C2 cell morphology and cell viability in a temperature and time-dependent manner.

### 2.2. HS Activates a TLR4-Mediated Inflammatory Environment in Cardiomyocytes

The H9C2 cells were cultured to approximately 75~80%. The control group was placed in an incubator at 37 °C. The HS group was placed in an incubator at 43 °C for 2 h for high temperature intervention, and then placed at 37 °C for recovery at different times (0 h, 3 h, 6 h, and 12 h). The schematic diagram of the experimental scheme is shown in Figure 2A. Compared with the control group, TLR4 expression was significantly increased at 3 h after recovery from heat exposure (*p* < 0.01) in the HS group (Figure 2B). Western blotting showed that, compared with the control group, the expressions of NF-κB and p53 also significantly increased at 3h recovery after thermal exposure in the HS group (Figure 2C), and the results were statistically significant (*p* < 0.01). At the mRNA level, TLR4, NF-κB, and p53 mRNA levels were significantly increased after 3 h recovery from heat exposure (*p* < 0.01) (Figure 2D). Meanwhile, the levels of inflammatory factors TNF-α, IL-6, and IL-1β in H9C2 cells were detected and significantly increased after 3 h recovery from heat exposure (*p* < 0.001) (Figure 2E). In conclusion, TLR4 significantly expressed in H9C2 cells after 3 h recovery from heat exposure, and the TLR4/NF-κB signaling pathway and the inflammatory environment were activated.

### 2.3. Ferroptosis of Cardiomyocytes Induced by HS Can Be Inhibited by Liproxstatin-1

The H9C2 cells were thermally exposed at 43 °C for 2 h and recovered for 3 h. To confirm the occurrence of ferroptosis due to heat exposure, we added the ferroptosis inhibitor, liproxstatin-1. The results showed that, compared with the control group, there was no significant difference in the expression of SLC7A11 and GPX4 in the liproxstatin-1 group, and the expression of SLC7A11 (*p* < 0.01) and GPX4 (*p* < 0.05) in the HS group significantly reduced. Compared with the HS group, SLC7A11 and GPX4 contents in the HS+Liproxstatin-1 group significantly increased (*p* < 0.05) (Figure 3A). GSH and MDA content were detected, and the results showed that, compared with the control group, the GSH and MDA contents in the liproxstatin-1 group showed no significant changes. The GSH contents (*p* < 0.01) in the HS group significantly decreased, while the MDA (*p* < 0.05) contents significantly increased. Compared with the HS group, the content of GSH in HS+Liproxstatin-1 group was increased, while the content of MDA was decreased(Figure 3B). To further confirm our hypothesis, we added erastin, a ferroptosis inducer. The mitochondria of H9C2 cardiomyocytes in each group were observed using transmission electron microscopy (TEM). The results showed that, compared with the control group, mitochondria in the liproxstatin-1 group had no obvious changes, while mitochondria in the HS group became smaller and membrane density increased. The mitochondrial changes in the erastin group were consistent with those in the HS group. The mitochondrial morphological structure in the HS+Liproxstatin-1 group was improved compared with that in the HS group (Figure 3C). ROS and Fe^2+^ content were detected, and the results showed that, compared with the control group, the ROS and Fe^2+^ contents in the liproxstatin-1 group showed no significant changes, and in the HS group were significantly increased. The changes in the erastin group were the same as those in the HS group. Compared with the HS group, the ROS, and Fe^2+^ contents in the HS+Liproxstatin-1 group were decreased (Figure 3D,E). In summary, HS may induce ferroptosis in H9C2 cells, which was consistent with the effect of erastin on H9C2 cells and was ameliorated by the ferroptosis inhibitor liproxstatin-1.

### 2.4. Inhibition of TLR4 Expression Can Improve Cardiomyocyte Injury Induced by HS

To further verify whether inhibition of TLR4 improved cell damage induced by HS, normal H9C2 cells were pretreated with different concentrations of TAK-242. Results showed that low concentrations (5, 25, and 50 μM) of TAK-242 had no effect on cell viability. High concentrations of TAK-242 (100, 500, and 1000 μM) reduced cell viability in a dose-dependent manner (*p* < 0.01) (Figure 4A). We pretreated H9C2 cells with low concentrations (5, 25, and 50 μM) of TAK-242, and placed them in an incubator at 43 °C for 2 h, and then in an incubator at 37 °C for 3 h. The results showed that 50 μM of TAK-242 significantly improved the viability of H9C2 cells (*p* < 0.05) (Figure 4B) and down-regulated HS-induced high expression of TLR4 (*p* < 0.01) (Figure 4C).

### 2.5. Inhibition of TLR4 Down-Regulates the Inflammatory Environment of Cardiomyocytes Induced by HS

To investigate the effects of HS on TLR4/NF-κB signaling pathway and pro-inflammatory factors, we used Western blotting, RT-PCR, and immunofluorescence to detect relevant indicators. The results showed that, compared with the control group, the expressions of TLR4, NF-κB, and p53 were not significantly different in the TAK-242 group but significantly increased in the HS group (*p* < 0.01). Compared with the HS group, the expressions of TLR4 (*p* < 0.05), NF-κB (*p* < 0.05), and p53 (*p* < 0.01) were significantly decreased in the HS + TAK-242 group (Figure 5A,B,D). ELISA results showed that, compared with the control group, the levels of TNF-α, IL-6, and IL-1β in the TAK-242 group were not significantly changed, but the levels of TNF-α, IL-6, and IL-1β in the H9C2 cells in the HS group were significantly increased (*p* < 0.001). Compared with the HS group, the levels of TNF-α, IL-6, and IL-1β were significantly decreased in the HS+TAK-242 group (*p* < 0.01) (Figure 5C). In conclusion, inhibition of TLR4 may down-regulate the TLR4/NF-κB signaling pathway and inflammatory factors in cardiomyocytes under HS conditions.

### 2.6. Inhibition of TLR4 Alleviates Ferroptosis of Cardiomyocytes Induced by HS

To explore the role of TLR4 in cardiocyte ferroptosis due to HS, H9C2 cells were pretreated with TAK-242 and ferroptosis-related indicators were detected by Western blotting, RT-qPCR, and immunofluorescence assay. The results were as expected and compared with the control group. The expressions of SLC7A11 and GPX4 in the TAK-242 group were not significantly different, while the expressions of SLC7A11 (*p* < 0.05) and GPX4 (*p* < 0.01) in the HS group were significantly reduced. The expressions of SLC7A11 (*p* < 0.05) and GPX4 (*p* < 0.01) in the HS + TAK-242 group were significantly increased compared with those in the HS group (Figure 6A–C). The TEM results showed that the mitochondrial morphology and structure of the TAK-242 group did not change significantly compared with the control group. Furthermore, the mitochondria of the HS group became smaller and the membrane density increased, and the mitochondria of the HS + TAK-242 group were improved compared with that of the HS group (Figure 6D). Compared with the control group, the GSH, MDA, ROS, and Fe^2+^ contents in the TAK-242 group were not significantly changed, while the GSH content in the HS group was significantly decreased (*p* < 0.01) and the MDA (*p* < 0.01), ROS, and Fe^2+^ contents were significantly increased. Compared with the HS group, the GSH (*p* < 0.05) content was significantly increased and MDA (*p* < 0.01), ROS, and Fe^2+^ contents in the HS + TAK-242 group were decreased (Figure 6E–G). In conclusion, inhibition of TLR4 may reverse ferroptosis injury to cardiomyocytes in HS conditions.

### 2.7. Inhibition of TLR4/NF-κB Signaling Pathway Alleviates the Inflammatory Environment and Ferroptosis of Cardiomyocytes Induced by HS

To investigate how inhibition of TLR4 further affects ferroptosis in cardiocytes caused by HS, we pretreated H9C2 cells with NF-κB inhibitor PDTC to detect TLR4 and its downstream NF-κB and inflammatory factors, as well as to detect ferroptosis-related indicators. RT-qPCR results showed that compared with the control group, the mRNA expressions of TLR4, NF-κB, p53, SLC7A11 and GPX4 in PDTC group were not significantly changed, while mRNA expressions of TLR4, NF-κB and p53 were significantly increased (*p* < 0.01), SLC7A11 and GPX4 were significantly decreased (*p* < 0.01) in the HS group. TLR4 in the HS + PDTC group was not significantly different from that in the HS group (*p* > 0.05). The mRNA levels of NF-κB and p53 were decreased, while the mRNA levels of SLC7A11 and GPX4 were increased (*p* < 0.01) (Figure 7A). Compared with the control group, the contents of TNF-α, IL-6, IL-1β, GSH, MDA, ROS, and Fe^2+^ in the PDTC group were not significantly different (*p* > 0.05). The contents of TNF-α, IL-6, IL-1β, MDA, ROS and Fe^2+^ in the HS group were significantly increased, and GSH was significantly decreased (*p* < 0.01). Compared with HS group, the contents of TNF-α, IL-6, IL-1β, MDA, ROS and Fe^2+^ in HS+PDTC group were significantly decreased, while the contents of GSH were increased (*p* < 0.001, *p* < 0.05), and the results were consistent with those of the HS + TAK-242 group (Figure 7B–E). In conclusion, inhibition of TLR4/NF-κB signaling pathway may down-regulate HS-induced inflammatory response and ferroptosis in H9C2 cells.

## 3. Discussion

The occurrence of HS is often accompanied by a systemic inflammatory response [15], which also plays a key role in the injury of myocardial tissue. Previous studies have shown that inhibiting the inflammatory response is an important means to treat myocardial injury caused by HS [16]. Lin et al. found that melatonin may protect against HS-induced myocardial injury in male rats by mitigating oxidative stress and inflammation [17]. At present, a large number of studies on the protective effect of HS-induced myocardial injury are still limited to down-regulating inflammatory response or oxidative stress, and experiments show that this does not effectively improve HS-induced myocardial injury, suggesting that there may be other injury mechanisms. Clinical data also showing poor prognosis in patients treated with anti-inflammatory therapy suggest that the damage to the myocardium of patients with HS may be delayed and may be accompanied by other mechanisms of injury. Therefore, the post-HS recovery process has attracted our attention. Increasing evidence shows that TLR4 is an important target for the treatment of cardiovascular related diseases [18]. However, there are few reports on the mechanism of TLR4 in HS-induced myocardial injury. Our research group [12] previously found that inhibition of TLR4 can improve cardiac dysfunction in rats with HS. However, the mechanism of TLR4-induced myocardial injury in HS remains unclear. Through investigation conducted at the cellular level, this study showed that TLR4 expression was significantly increased when H9C2 cells were exposed to high temperature in a 43 °C incubator for 2 h and then recovered in a 37 °C incubator for 3 h [19]. Based on this result, we investigated the expression of TLR4/NF-κB pathway protein and p53 at the protein and mRNA levels, and the contents of inflammatory factors TNF-α, IL-6, and IL-1β. The results also confirmed our speculation that the inflammatory environment may be significantly activated under HS conditions.

Ferroptosis is a newly discovered type of iron-dependent programmed cell death [6]. The light chain subunit SLC7A11 and the heavy chain subunit SLC3A2 of the cysteine/glutamate anti-transporter form the system X_c_^−^ [20] and are responsible for basic transport activity in the system X_c_^−^ [21]. Down-regulation of SLC7A11 can indirectly inhibit the activity of GPX4 by inhibiting the cysteine metabolism pathway, resulting in decreased cystine levels in cells and the depletion of GSH biosynthesis, thus leading to the accumulation of lipid peroxides and ultimately inducing ferroptosis in cells [22]. Numerous studies have found that in various experimental models, investigating cardiovascular diseases such as myocardial ischemia reperfusion, myocardial infarction, and heart failure, as well as heart transplantation, the molecular and metabolic regulatory mechanisms of ferroptosis play a key role and can be significantly improved by inhibiting ferroptosis [23,24,25]. Erastin can reduce intracellular cysteine and GSH by constraining system X_c_^−^, thereby inhibiting GPX4 from clearing ROS, changing mitochondrial morphology, and inducing ferroptosis [26]. However, whether ferroptosis plays an important role in HS-induced myocardial cell injury has not been reported. Our previous in vivo studies found that HS caused decreased expression of SLC7A11 and GPX4 in myocardial tissue [12], which is consistent with our current in vitro results. Further study showed that HS could significantly reduce GSH content and increased MDA, ROS, and Fe^2+^ content in H9C2 cells, and lead to mitochondrial shrinkage and membrane density increases. These changes were consistent with the effects of erastin found on H9C2 cells, which can be improved with liproxstatin-1.

TLR4 is a pattern recognition receptor, which can regulate various cell death modes in myocardial tissue such as inflammation, apoptosis, autophagy, and ferroptosis [27,28,29]. Of concern, TLR4 knockout slowed autophagy and ferroptosis of activated cardiomyocytes in heart failure rats [28]. Heart failure is an important complication of HS [30]. The TLR4/NF-κB signaling pathway is an important inflammatory signaling pathway [31] involved in the pathophysiological processes of cardiovascular diseases such as cardiac ischemia, ventricular remodeling, fibrosis, and heart failure [32,33,34,35]. It has been reported that in older adult patients with chronic heart failure complicated with lung infection, the activation degree of the TLR4/NF-κB signaling pathway and the expression levels of inflammatory factors such as IL-4, IL-6, and TNF-α are increased, and that the inflammatory response and cardiac function damage are significantly aggravated [36]. Blocking the TLR4/NF-κB signaling pathway may help reduce myocardial injury and improve cardiac function after a coronary micro-embolism [37]. Whether the TLR4/NF-κB signaling pathway mediates ferroptosis in HS conditions is unclear. Our experimental results showed that, by inhibiting TLR4, the expression levels of TLR4, NF-κB, and p53 in H9C2 cells affected by HS at both protein and mRNA levels and the inflammatory environment mediated by inflammatory factors TNF-α, IL-6, and IL-1β were decreased. However, the potential link between HS-induced inflammation and ferroptosis remains to be explored. As a transcription factor, p53, one of the downstream molecules of TLR4, can act on SLC7A11 and inhibit the expression of SLC7A11 and GPX4, thereby inducing ferroptosis. Studies have shown that p53-mediated intervention of NF-κB induces apoptosis, autophagy, and cell death [13]. This suggests that HS may induce ferroptosis through the TLR4/NF-κB/p53 signaling pathway (see Figure 8 schematic diagram of the research hypothesis for the specific process). To verify our hypothesis, we inhibited TLR4 and NF-κB, respectively, under HS conditions and detected the inflammatory response and ferroptosis-related indicators. The results showed that inhibition of TLR4 and NF-κB down-regulated p53 expression and improved ferroptosis-related indicators and the inflammatory environment. Taken together, these results suggest, for the first time, that inhibition of TLR4 may improve ferroptosis in H9C2 cardiomyocytes induced by HS, and that inhibition of the TLR4/NF-κB signaling pathway may exert anti-inflammatory and anti-ferroptosis effects at the cellular level.

This study has some limitations. First, this study did not undertake an in-depth analysis of possible mechanisms. We only proposed that ferroptosis may be involved in the cell death pathway caused by HS and revealed that it may induce ferroptosis through the TLR4/NF-κB/p53 pathway. However, the specific interaction between NF-κB and p53 remains to be investigated, which will be a focus of our research group in future. Another limitation is that we used H9C2 rat cardiomyocyte lines instead of primary rat cardiomyocytes. H9C2 cells are the most used substitute for rat cardiomyocytes, but there are still some differences between these and primary rat cardiomyocytes, which may affect the generalizability of our results.

## 4. Materials and Methods

### 4.1. Cell Culture and Intervention

An H9C2 rat myocardial cell line was purchased from the American Type Culture Collection (ATCC). H9C2 rat cardiomyocytes were cultured in an incubator (37 °C, 5% CO_2_) with DMEM (HyClone, Logan, UT, USA) containing 10% FBS (BI, Israel). The H9C2 cells were treated at different temperatures (37 °C, 41 °C, 43 °C, and 45 °C) for 2 h, and the intervention temperature was screened. The H9C2 cells were treated at a high temperature of 43 °C for different durations (2 h, 3 h, 4 h, and 5 h), to screen out the duration of heat exposure. According to the experimental design, the experiment was divided into four parts. The first part comprised a control group and an HS group. The control group was cultured in an incubator at 37 °C, and the HS group was treated at 43 °C for 2h and then placed in the incubator at 37 °C for recovery at different time points (0 h, 3 h, 6 h, and 12 h; MCE, New Jersey, NJ, USA) to screen out the appropriate recovery time. The second part comprised a control group, a liproxstatin-1 (25 μmol/L; MCE, New Jersey, NJ, USA) group, an HS group, an HS + liproxstatin-1 group, and an erastin (30 μmol/L; MCE, New Jersey, NJ, USA) group. The third part comprised a control group, a TAK-242 (50 μmol/L; MCE, New Jersey, NJ, USA) group, an HS group, and an HS + TAK-242 group. The fourth part comprised a control group, a PDTC (100 μmol/L; MCE, New Jersey, NJ, USA) group, an HS group, an HS + PDTC group, and an HS + TAK-242 group. TAK-242, a TLR4 inhibitor, was dissolved and diluted to the required concentration (5, 25, 50, 100, 500, and 1000 μM). It was then added into 96 wells and incubated at 37 °C or 43 °C for 2 h to screen out the appropriate concentration of TAK-242. Liproxstatin-1 is an ferroptosis inhibitor, erastin is an ferroptosis inducer, and PDTC is an NF-κB inhibitor when used in the appropriate concentrations, according to the manufacturer’s instructions.

### 4.2. Cell Viability Assays

Cell viability was detected using a cell counting kit-8 (CCK-8) assay (UE, Shanghai, China). The H9C2 cells (5 × 10^3^ cells/mL) were treated with various TAK-242 concentrations for 24 h or at different temperatures for 2 h, after which 10μL volume of CCK-8 reagent (Zeta Life, Atlanta, GA, USA) was added and incubation was undertaken at 37 °C for 1 h. After incubation, OD values were detected at 450 nm using a microplate reader (Thermo Fisher Scientific, Waltham, MA, USA).

### 4.3. Western Blotting

The H9C2 cells were collected in a 25 cm culture flask and cleaved to obtain total protein using a pre-cooled RIPA lysis buffer containing 1 mM PMSF. Total protein concentration was determined using a BCA protein detection kit (KeyGEN, Jiangsu, China). The same amount of protein was separated by 10% SDS-PAGE gel and transferred to a polyvinylidene fluoride (PVDF) membrane. The membrane was closed with rapid sealing solution at room temperature for 30 min and washed with PBST. TLR4 (1:1000; Santa Cruz, CA, USA), NF-κB (1:2000; Abbkine, Wuhan, China), SLC7A11 (1:500; Affinity, PA, USA), p53 (1:1000; Biolead, Beijing, China), GPX4 (1:1000; Abmart, Shanghai, China), and GAPDH (1:5000; Elabscience, Wuhan, China) were incubated at 4 °C overnight and washed with PBST. Goat anti-rabbit IgG (1:5000; Elabscience, Wuhan, China) or goat anti-mouse IgG (1:5000; Biogot, Nanjing, China) labeled with horseradish peroxidase was added and incubated at room temperature for 1 h and washed with PBST. Following thrice washing, the membranes were detected using a Western blot ECL kit (EpiZyme, Shanghai, China).

### 4.4. RNA Extraction and Analysis

Total RNA was extracted from the H9C2 cells using an RNA extraction kit (Tiangen, Beijing, China). Total RNA was reverse-transcribed into cDNA using a PrimeScriPt RT Master Mix (TaKaRa, Osaka, Japan) according to the manufacturer’s instructions along with a SYBR PremiexExTaq kit (TaKaRa, Osaka, Japan) to target cDNA qPCR augmentation. GAPDH was the internal parameter. The PCR primer sequence (Servicebio, Wuhan, China) is as follows: TLR4 forward:5′-CCAGGTGTGAAATTGAGACAATTG-3′, TLR4 reverse: 3′-AAGCTGTCCAATATGGAAACCC-5′.NF-κB forward:5′-CAGATACCACTAAGACGCACCC-3′, NF-κB reverse: 3′-CTCCAGGTCTCGCTTCTTCACA-5′.p53 forward: 5′-GGAGGATTCACAGTCGGATATG-3′, p53 reverse: 3′-TGAGAAGGGACGGAAGATGAC-5′.SLC7A11 forward: 5′-TATGCTGAATTGGGTACGAGC-3′, SLC7A11 reverse: 3′-TATTACCAGCAGTTCCACCCA-5′.GPX4 forward: 5′-AGGCAGGAGCCAGGAAGTAATC-3′.GPX4 reverse: 3′-ACCACGCAGCCGTTCTTATC-5′.GAPDH forward: 5′-CTGGAGAAACCTGCCAAGTATG-3′, GAPDH reverse: 3′-GGTGGAAGAATGGGAGTTGCT-5′.

### 4.5. Immunofluorescence

The H9C2 cells were inoculated into a confocal dish, fixed with 4% paraformaldehyde for 30 min after intervention, drilled with 0.5% TritonX-100 for 10 min, then closed with goat serum at room temperature for 30 min.α-actinin (1:100; Proteintech, Wuhan, China), TLR4 (1:500; Servicebio, Wuhan, China), NF-κB (1:100; Abmart, Shanghai, China), SLC7A11 (1:100; Affinity, PA, USA), and GPX4 (1:100; Abmart, Shanghai, China) primary antibody were added and incubated at 4 °C overnight. After washing with PBS, either goat anti-rabbit IgG (1:50; ZSGB-BIO, Beijing, China) or goat anti-mouse IgG (1:50; ZSGB-BIO, Beijing, China) was incubated at room temperature for 60 min. Nuclear staining was performed using DAPI (Solarbio, Beijing, China) and finally observed under a confocal microscope.

### 4.6. Cytokine Measurements

The contents of TNF-α, IL-6, and IL-1β in H9C2 rat cardiomyocytes were determined using a high-sensitivity ELISA kit (Multi Sciences Biotech, Hangzhou, China). A total of 100 μL cell culture supernatant was added into the well plate of the kit. After incubation and washing, diluted TNF-α, IL-6, and IL-1β antibodies were added (1:100). After washing, 100 μL diluted horseradish peroxidase-labeled chain enzyme avidin (1:100) was added to each well. After washing, 100 μL color-producing substrate TMB was added to each well and incubated in the dark for 10 min, after which 100 μL termination solution was added to each well. OD values of 450 nm and 570 nm were detected using the microplate reader.

### 4.7. Determination of ROS Levels

The ROS content of the H9C2 cells was determined using a ROS detection kit (UE, Suzhou, China). The H9C2 cells were inoculated into 12-well plates, and the cells grew to 85% before intervention modeling. After cleaning with PBS, 1 mL DCFH-DA working solution of 10μM was added to each well, and the cells were incubated in a cell incubator at 37°C for 30 min in the dark. The cells were washed twice with serum-free culture solution. Finally, they were observed and photographed under a fluorescence microscope.

### 4.8. Transmission Electron Microscopy (TEM)

Intervention modeling was performed when the H9C2 cells reached 85%. After centrifugation at 2000× *g* for 5 min, the cells were collected, 2.5% glutaraldehyde was added, and they were fixed at room temperature for 30 min away from light. The samples of each group were dehydrated, embedded, and cut into ultrathin slices. Finally, TEM was used to observe and collect images under a magnification of 1200×, 2000×, and 10,000×.

### 4.9. Measurement of Malonaldehyde (MDA) and Glutathione (GSH) Levels

The H9C2 cells were collected after intervention for ultrasonic fragmentation. A total of 300 μL MDA working solution (Solarbio, Beijing, China) and 100μL sample were mixed, and kept warm in a 100 °C water bath for 60 min, cooled in an ice bath, centrifuged at 10,000× *g* at room temperature for 10 min, after which 200μL absorbing supernatant was added to the 96-well plate. The absorbance was measured at 532 nm and 600 nm. The GSH content in the H9C2 cells was determined using a GSH detection kit (Abbkine, Wuhan, China). The H9C2 cells were collected after the intervention, freeze-thawed three times, and centrifuged at 4 °C at 8000× *g* for 10 min. After the supernatant was removed, they were added to the 96-well plate according to the manufacturer’s instructions and mixed well. They were then incubated at room temperature and away from light for 2 min, with the absorbance value recorded at 412 nm.

### 4.10. Iron Content Assay

The H9C2 cells were inoculated in a confocal dish and cultured overnight in a 5% CO_2_ incubator at 37 °C. After the intervention, FerroOrange (DOJINDO, Kumamoto Ken, Japan) working liquid with a concentration of 1 mol/L was added to the confocal dish and cultured for 30 min in an incubator at 37 °C and 5% CO_2_. Finally, observations and photographs were taken directly under confocal microscopy.

### 4.11. Statistical Analysis

All statistical analyses was performed using SPSS 19.0 software and the statistical graphs were processed using Graphpad Prism8. Data from at least three independent experiments were used for statistical analysis and presented as mean ± standard deviation (SD). Comparisons between the two groups were conducted using a student’s *t* test for normal distribution data, and comparisons between multiple groups were conducted using one-way ANOVA analysis. A *p*-value of < 0.05 was considered a statistically significant difference.

## 5. Conclusions

In this study, HS-induced changes in H9C2 cardiomyocyte morphology and cell viability were found to occur in a time- and temperature-dependent manner. Moreover, our results demonstrated for the first time that HS can activate an TLR4/NF-κB-induced ferroptosis. Inhibition of the TLR4/NF-κB pathway can improve H9C2 cell viability and down-regulate the inflammatory environment and ferroptosis.

## Figures and Tables

**Figure 1 molecules-28-02297-f001:**
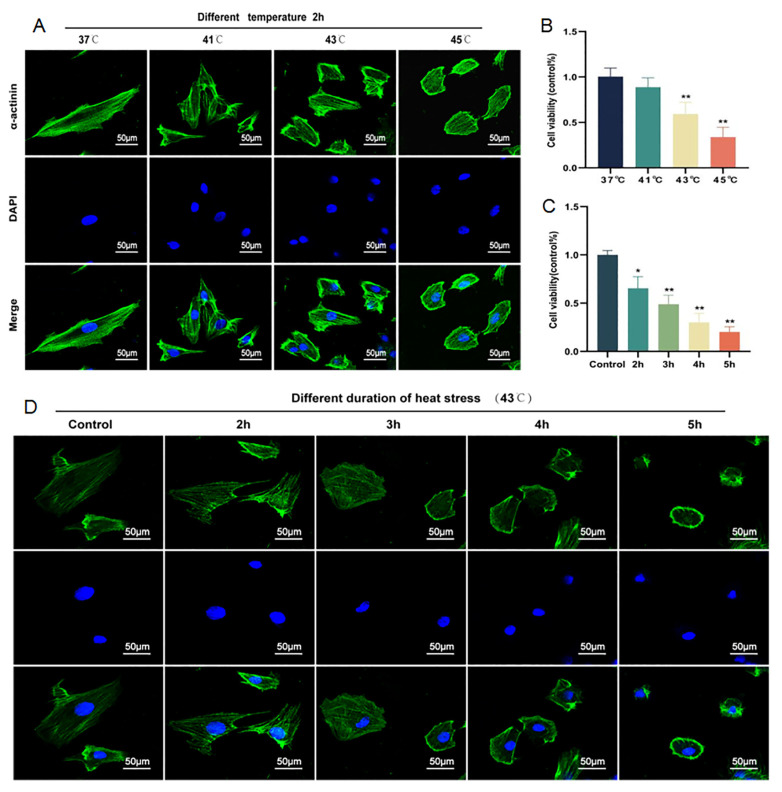
Morphological and viability changes in H9C2 cells. Cell morphology of H9C2 cells at different temperatures (37 °C, 41 °C, 43 °C and 45 °C) for 2 h. Scale bars = 50 µm. (**A**). Cell viability of H9C2 cells at different temperatures for 2 h via a CCK-8 assay (**B**). Cell viability of H9C2 cells at 43 °C for different duration (2 h, 3 h, 4 h, and 5 h) via a CCK-8 assay (**C**). Cell morphology of H9C2 cells at 43 °C for a different duration. Scale bars = 50 µm. (**D**). The staining was performed by FITC-conjugated α-actinin and DAPI and observed by confocal fluorescence microscopy. The results were expressed as the mean ± SD deviation of four or six independent experiments, * *p* < 0.05, ** *p* < 0.01 vs. the control group.

**Figure 2 molecules-28-02297-f002:**
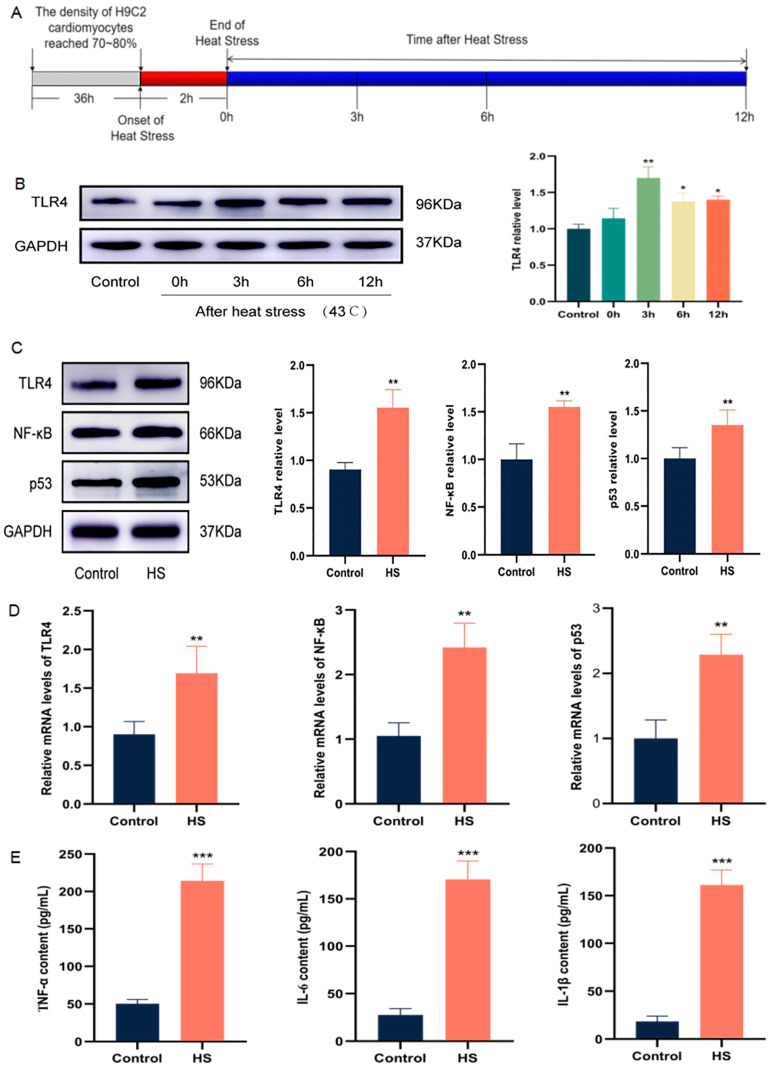
HS activates TLR4/NF-κB signaling pathway and inflammatory environment. The in vitro experimental protocol (**A**). TLR4 expression levels at different time points after heat exposure (**B**). The protein levels of TLR4, NF-κB, and p53 after 3 h recovery from heat exposure (**C**). The mRNA levels of TLR4, NF-κB, and p53 after 3 h recovery from heat exposure (**D**). The levels of TNF-α, IL-6, and IL-1β after 3 h recovery from heat exposure (**E**). The results were expressed as the mean ± SD deviation of three independent experiments, * *p* < 0.05, ** *p* < 0.01, *** *p* < 0.001 vs. the control group.

**Figure 3 molecules-28-02297-f003:**
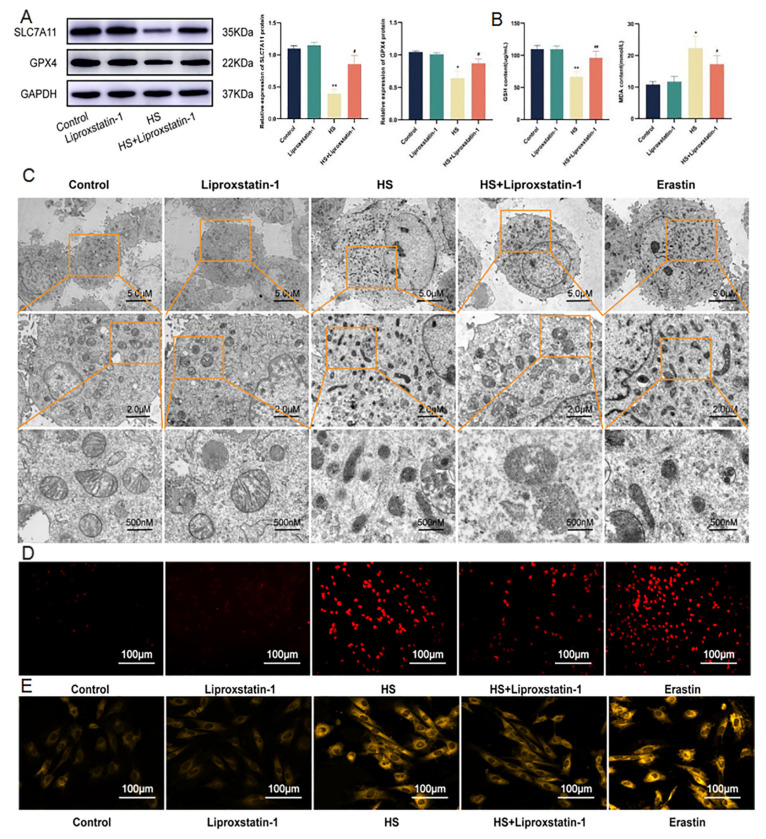
Liproxstatin-1 ameliorates ferroptosis induced by HS. The protein expression of SLC7A11 and GPX4 in H9C2 cells (**A**). GSH and MDA contents in H9C2 cells (**B**). Image of mitochondrial structure of H9C2 cells at 1200×, 2000×, or 10,000× magnification croscope. Scale bars = 5 µm, 2 µm and 500 nm (**C**). ROS level in H9C2 cells. Scale bars = 100 µm (**D**). Fe^2+^ level in H9C2 cells. Scale bars = 100 µm (**E**). The results were expressed as the mean ± SD deviation of three independent experiments. * *p* < 0.05, ** *p* < 0.01 vs. the control group; ^#^
*p* < 0.05, ^##^
*p* < 0.01 vs. the HS group.

**Figure 4 molecules-28-02297-f004:**
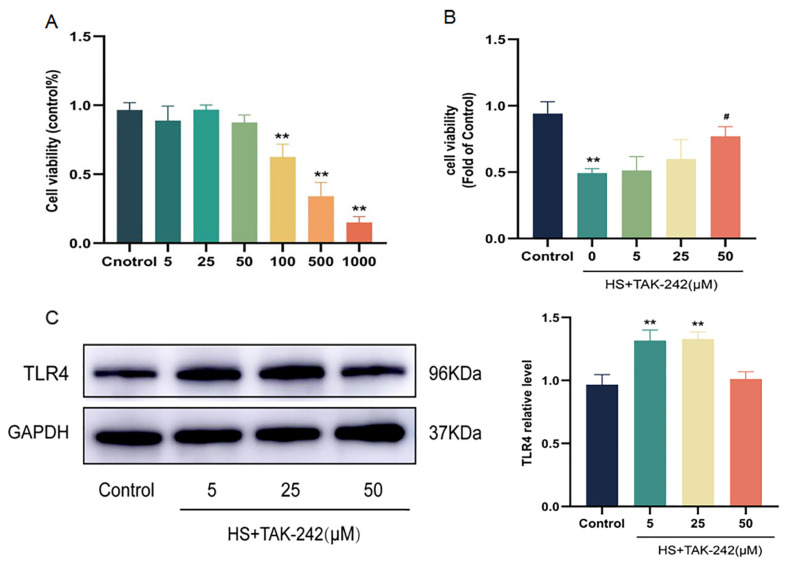
Effects of TAK-242 pretreatment on H9C2 cell viability and TLR4 expression level. Effect of different concentrations of TAK-242 on H9C2 cell viability (**A**). Effect of low concentration TAK-242 on H9C2 cell viability after heat exposure (**B**). Effect of low concentration of TAK-242 on TLR4 expression in H9C2 cells after heat exposure (**C**). The results were expressed as the mean ± SD deviation of three independent experiments, ** *p* < 0.01 vs. the control group; ^#^
*p* < 0.05 vs. the HS group.

**Figure 5 molecules-28-02297-f005:**
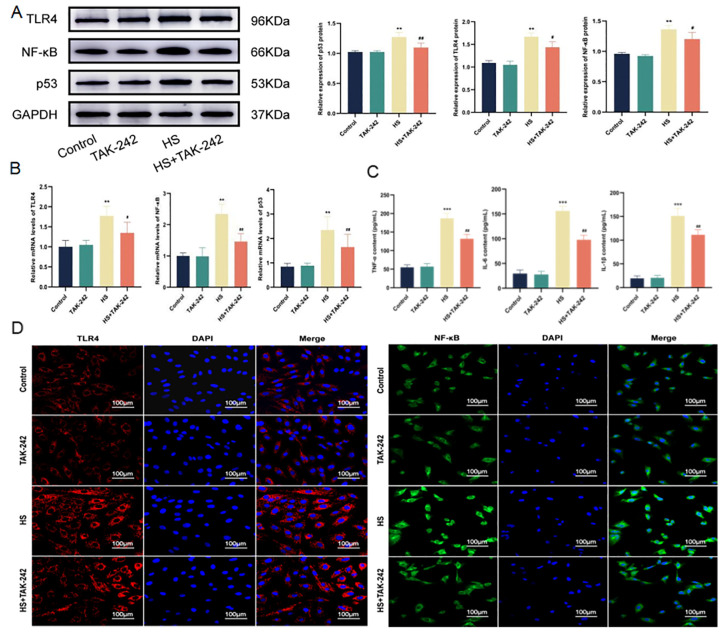
TAK-242 inhibits the HS-induced inflammatory environment. The protein levels of TLR4, NF-κB, and p53 in H9C2 cells were determined using Western blotting (**A**). The mRNA levels of TLR4, NF-κB, and p53 in H9C2 cells were determined using RT-qPCR (**B**). TNF-α, IL-6, and IL-1β contents in H9C2 cells (**C**). Fluorescence images of H9C2 cells stained with DAPI, anti-TLR4 antibody, and anti-NF-κB antibody. Scale bars = 100µm (**D**). The results were expressed as the mean ± SD deviation of three independent experiments, ** *p* < 0.01, *** *p* < 0.001 vs. the control group; ^#^
*p* < 0.05, ^##^
*p* <0.01 vs. the HS group.

**Figure 6 molecules-28-02297-f006:**
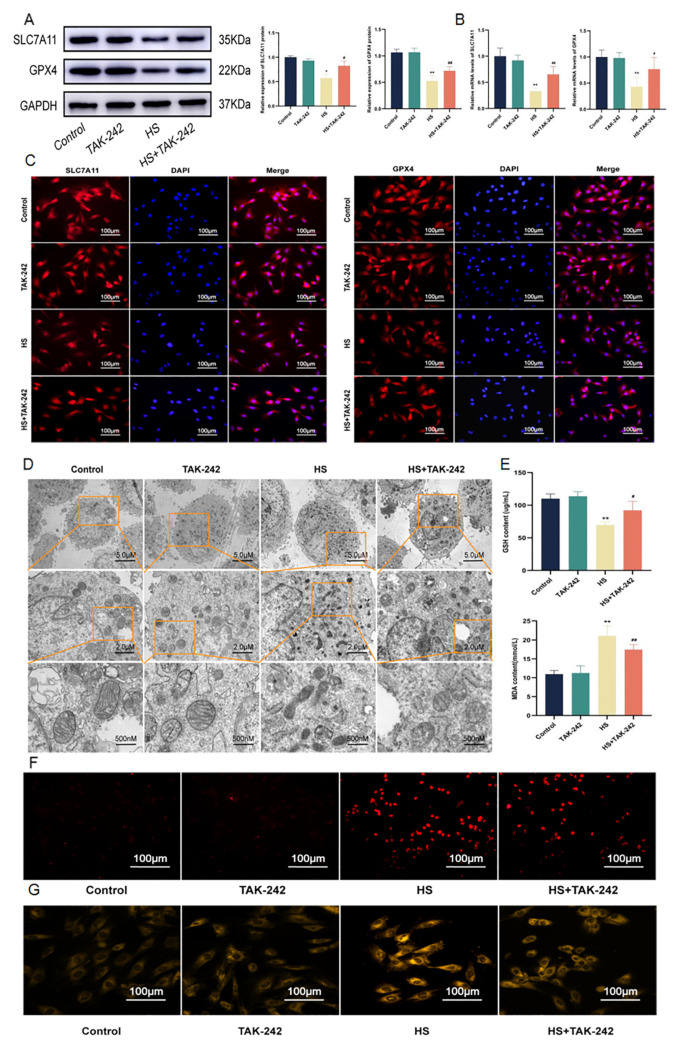
TAK-242 improved HS-induced ferroptosis. The protein levels of SLC7A11 and GPX4 in H9C2 cells were determined using Western blotting (**A**). The mRNA levels of SLC7A11 and GPX4 in H9C2 cells were determined using RT-qPCR (**B**). Fluorescence images of H9C2 cells stained with DAPI, anti-GPX4 antibody, and anti-SLC7A11 antibody. Scale bars = 100 µm (**C**). Image of mitochondrial structure of H9C2 cells at 1200×, 2000×, or 10,000× magnification. Scale bars = 5 µm, 2 µm and 500 nm (**D**). GSH and MDA levels in H9C2 cells (**E**). ROS level in H9C2 cells. Scale bars = 100 µm (**F**). Fe^2+^ level in H9C2 cells. Scale bars = 100 µm (**G**). The results were expressed as the mean ± SD deviation of three independent experiments. * *p* < 0.05, ** *p* < 0.01 vs. the control group; ^#^
*p* < 0.05, ^##^
*p* < 0.01 vs. the HS group.

**Figure 7 molecules-28-02297-f007:**
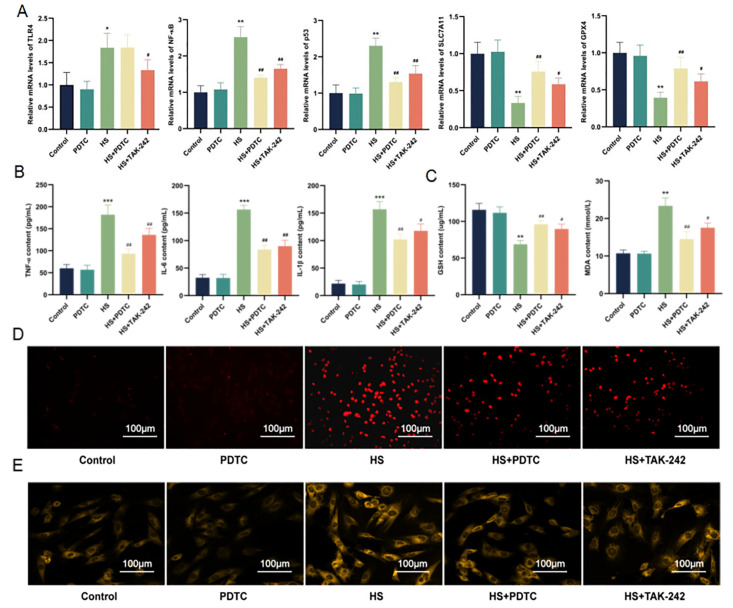
Inhibition of TLR4 attenuates ferroptosis by suppressing TLR4/NF-κB signaling. The mRNA levels of TLR4, NF-κB, p53, SLC7A11 and GPX4 in H9C2 cells were determined using RT-qPCR (**A**). TNF-α, IL-6, and IL-1β contents in H9C2 cells (**B**). GSH and MDA contents in H9C2 cells (**C**). ROS level in H9C2 cells. Scale bars = 100 µm (**D**). Fe^2+^ level in H9C2 cells. Scale bars = 100 µm (**E**). The results were expressed as the mean ± SD deviation of three independent experiments. * *p* < 0.05, ** *p* < 0.01, *** *p* < 0.001 vs. the control group; ^#^
*p* < 0.05, ^##^
*p* < 0.01 vs. the HS group.

**Figure 8 molecules-28-02297-f008:**
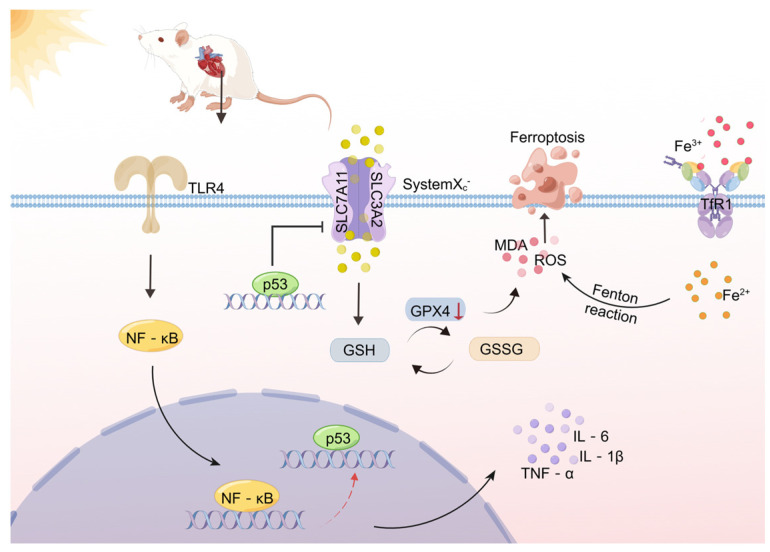
A schematic diagram of the research hypothesis concerning the inflammatory response and ferroptosis of cardiomyocytes caused by HS (by Figdraw).

## Data Availability

The data that support the findings of this study are available from the corresponding author upon reasonable request.

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
