# Peer review of "Inhibition of TLR4 Alleviates Heat Stroke-Induced Cardiomyocyte Injury by Down-Regulating Inflammation and Ferroptosis"

_molecules, 2023, doi:10.3390/molecules28052297_

Round 1

Reviewer 1 Report

In this study, a series of experiments and phenotype observations demonstrated that HS activated TLR4/NF-κB induced inflammatory response and ferroptosis in a time- and temperature-dependent manner, thereby affecting the morphological changes of H9C2 cardiomyocytes. The whole experiment design is complete and worthy of recommendation for publication. However, some minor problems still need the author's attention.

1. Please specify the specific time and temperature indicated in Figure 1B and 1C.

2. The increased expression of TLR4, NF-κB and inflammatory factors does not necessarily indicate the interaction between them, so please carefully state the results (L128-130).

3. Please pay attention to the spaces between words, such as L198, L199, L201, L208, L209, etc.

4. Please add corresponding picture numbers in Figure 4 and 5.

5. In Figure 4, compared with the control group, TLK4 expression level did not seem to change significantly at 50 uM of TAK-242, please further confirm the results (L181).

6. It is suggested to change "Figure 6A, B, C" to "Figure 6a-C".

7. Please supplement the full name of the abbreviation TEM in the manuscript.

8. The expression of "oC" is incorrect, please modify the whole manuscript, such as L356, L358 and L363, etc

9. The figure notes of each figures indicated that the data was shown as Mean±SEM, but it was indicated that the data was expressed as Mean±SD in the part of Materials and Methods. Please check which is correct.

Reviewer 2 Report

The level of originality of the paper is high. The literature review and proposed methodology are properly discussed and not compared to the previous studies about Inhibition of TLR4 Alleviates Heat Stroke-Induced Cardiomyocyte Injury by Down-Regulating Inflammation and Ferroptosis.

In this paper, authors used 35 sources, containing both historical and fundamental works, as well as the latest scientific research on this topic. But the literature review can be structured. The papers discussed many points of this study. Please, discuss these papers.

Development of Polymer Film Coatings with High Adhesion to Steel Alloys and High Wear Resistance. Polymer Composites, 41(7), 2875-2880. https://doi.org/10.1002/pc.25583

QoS-Ledger: Smart Contracts and Metaheuristic for Secure Quality-of-Service and Cost-Efficient Scheduling of Medical-Data Processing. Electronics, 10, 3083. https://doi.org/10.3390/electronics10243083

The introduction section has benefit from having a clearer structure of what to expect in the paper. Furthermore, the author(s) would benefit from being more concise in their writing, as much of the content was redundant and overemphasized. While it is good practice to assume the reader has no prior knowledge of the content, a topic and/or discussion does not need to be explained over and over again if it is stated both adequately and appropriately once.

Some conclusions contribute to the study of the problem. The author does not formulate the problem itself – it makes impossible to analyse the contribution of the paper. The aim or the question of the paper (or even the hypothesis of the author) are formulated.

Overall, it is very clear to grasp understanding of the manuscript and content in its current state. I strongly advise using hypothesis points to articulate and/or express material in scientific writing. Publication of this piece seems likely in any reputable scientific periodical after a correction in the writing of the manuscript.

Figure 2 is important to explore the specifics. Some conclusions can contribute to the study of the problem.

Authors need to add more details on the range of simulation considered in this work should be clearly outlined within the abstract. The current statements are vague and too general to get an idea of the work that have been accomplished.

Authors need to add more details on this particular works within citations [5-6].

The paper possesses a proper form of well-structured and readable technical language of the field and represents the expected knowledge of the journal`s readership.

There are minor errors in English, but this does not affect the general nature of the work. The current study brings many new to the existing literature or field. For one, the author(s) seem to have a good grasp of the current literature on their topic area (i.e., recent literature and seminal texts relevant to their study is not cited/referenced).

Reviewer 3 Report

In this article, Chen et al. investigated the role of ferroptosis and the associated mechanism in the context of heat shock (HS) induced cardiomyocyte Injury. They observed inhibition of TLR4 or NF-κB signaling pathway leads to down-regulation of inflammatory response and ferroptosis. The analysis has carefully performed some clarifications are needed to strengthen the rationale of the study and the conclusions. Comments are as follows:

Comments:

1. In the abstract and throughout the text, the authors mentioned that the inhibitors can improve the high or low expression of different genes. The authors should be more specific and scientific i.e., upon drug treatment what changes are occurring should be clearly mentioned (increase or decrease).

2. The abstract is very big and too many details are provided. It should be more concise outlining major findings.

3. For NF-κB blots, are the authors looking for p65? What is the effect on phospho-p65 since that documents the activation of NF-κB signaling? The authors should show the status of phospho-p65 upon HS and drug treatment.

4. What is the effect of combinatorial treatment of TLR4 and NF-κB inhibition? The authors should perform a combinatorial treatment to check whether that can further improve the condition.

Round 2

Reviewer 3 Report

The authors tried to address my concerns as much as possible. The article can be published now.